# Evolution of the interactions between GII.4 noroviruses and histo-blood group antigens: Insights from experimental and computational studies

Yu Liang[1], Wei Bu Wang[2], Jing Zhang[1], Jun Wei Hou[1], Fang Tang[1], Xue Feng Zhang[1], Li Fang Du[1], Ji Guo Su[2]*, Qi Ming Li[1]*

1 The Sixth Laboratory, National Vaccine and Serum Institute, Beijing, China, 2 Key Laboratory for Microstructural Material Physics of Hebei Province, College of Science, Yanshan University, Qinhuangdao, China

* jiguosu@ysu.edu.cn (JGS); liqiming189@163.com (QML)

**Data Availability Statement:** All relevant data are within the manuscript and its Supporting Information files.

## Abstract

Norovirus (NoV) is the major pathogen causing the outbreaks of the viral gastroenteritis across the world. Among the various genotypes of NoV, GII.4 is the most predominant over the past decades. GII.4 NoVs interact with the histo-blood group antigens (HBGAs) to invade the host cell, and it is believed that the receptor HBGAs may play important roles in selecting the predominate variants by the nature during the evolution of GII.4 NoVs. However, the evolution-induced changes in the HBGA-binding affinity for the GII.4 NoV variants and the mechanism behind the evolution of the NoV-HBGA interactions remain elusive. In the present work, the virus-like particles (VLPs) of the representative GII.4 NoV stains epidemic in the past decades were expressed by using the *Hansenula polymorpha yeast* expression platform constructed by our laboratory, and then the enzyme linked immunosorbent assay (ELISA)-based HBGA-binding assays as well as the molecular dynamics (MD) simulations combined with the molecular mechanics/generalized born surface area (MMGBSA) calculations were performed to investigate the interactions between various GII.4 strains and different types of HBGAs. The HBGA-binding assays show that for all the studied types of HBGAs, the evolution of GII.4 NoVs results in the increased NoV-HBGA binding affinities, where the early epidemic strains have the lower binding activity and the newly epidemic strains exhibit relative stronger binding intensity. Based on the MD simulation and MMGBSA calculation results, a physical mechanism that accounts for the increased HBGA-binding affinity was proposed. The evolution-involved residue mutations cause the conformational rearrangements of loop-2 (residues 390–396), which result in the narrowing of the receptor-binding pocket and thus tighten the binding of the receptor HBGAs. Our experimental and computational studies are helpful for better understanding the mechanism behind the evolution-induced increasing of HBGA-binding affinity, which may provide useful information for the drug and vaccine designs against GII.4 NoVs.

**Funding:** This work was supported by the grant from the Major Project of National Science and Technology of China on New Drug Creation and Development to Q.M.L. (2018ZX09739002). The funders had no role in study design, data collection and analysis, decision to publish, or preparation of the manuscript.

**Competing interests:** The authors have declared that no competing interests exist.

## Author summary

Human norovirus (NoV) has been recognized as the leading cause of the epidemic acute gastroenteritis worldwide and more than 50% acute gastroenteritis outbreaks are associated with NoVs. NoVs are highly infectious and may result in serious dehydration, malnutrition and even death, which severely threatens human health and brings heavy economic burden. NoVs are highly genetically diverse, in which the GII.4 genotype is the most predominant. The reported outbreaks of NoV infections have risen sharply from 2002, and it is suggested that the increasing NoV infections are attributed to the emergence of new strains with more infectiousness. GII.4 NoV evolves rapidly and on average every 2–3 years a new strain appears. It has been revealed that the histo-blood group antigens (HBGAs) serve as the recognition receptor for the GII.4 NoVs infecting the host cell, and the NoV-HBGA interactions may play an important role in selecting the predominate variants during the evolution of GII.4 NoVs. However, the molecular mechanism behind the evolution of the NoV-HBGA binding affinities is still not clear. In this work, the representative GII.4 NoV strains prevalent in the past decades were expressed, and the changes in the interactions between these strains and the receptor HBGAs were investigated by using the experimental measurements combined with computational simulations. Based on the experimental and computational results, a molecular mechanism that accounts for the increasing of the NoV-HBGA binding affinities during the evolution of GII.4 NoVs was proposed. Our studies are helpful for the understanding of the evolution mechanism of GII.4 NoVs and provide valuable information for the drug and vaccine designs against GII.4 NoVs.

## Introduction

Norovirus (NoV) is a highly infectious pathogen of human, which is responsible for most of the outbreaks of the viral gastroenteritis around the world. NoV causes illness in people of all ages, and it is especially more virulent for children and aged populations [1, 2]. NoV is highly genetically heterogenous and diverse, which can be grouped into seven genogroups, i.e., genogroup I (GI) to GVII [3, 4]. Among these genogroups, GI, GII and GIV can infect human, each of which can be further divided into many genotypes [4]. To date, NoV of GII.4 (genogroup II and genotype 4) is the most predominant genotype that accounts for the major outbreaks of NoV-involving diseases worldwide [1–6].

GII.4 NoV evolves rapidly and several different variants have been appeared. The pandemic of each variant dominates for some time, which is then substituted by a new emerging variant. It is believed that the US95/96 variant of GII.4 NoV is the first strain that causes the NoV pandemic in human [7, 8]. Subsequently, the Farmington Hills strain emerged to replace the US95/96 strain, where an additional residue was inserted into a surface loop around the receptor-binding site in the Farmington variant [7]. In 2004, the GII.4 variant Hunter was detected in Europe, Australia, and Asia, which spread worldwide [9–11]. In early 2006, the outbreaks of gastroenteritis in Australia, New Zealand and Europe were caused by the GII.4 variants termed as 2006a and 2006b strains [12]. After that, the variant 2008 was first detected in 2008 and subsequently resulted in NoV outbreaks all over the world from 2008 to 2009 [13, 14]. In October 2009, a new GII.4 variant termed New Orleans was detected from the samples of NoV outbreaks in the United States, which then predominated NoV spreads during 2009 to 2012 [15, 16]. After the prevalence of the variant New Orleans 2009, a new strain, Sydney 2012, began to be predominant in NoV circulation from the year of 2012 [17, 18]. Following that, the increase

in NoV outbreaks during 2016–2017 was found involving a new recombinant strain combining the GII.P16 polymerase with GII.4 Sydney 2012 termed as Sydney 2012 GII.P16/GII.4 strain [19, 20]. According to the above discussions, each GII.4 variant was kept prevalent for a period of about 2–3 years and then replaced by a new emerging strain. In addition, several studies have indicated that compared with the old variants, the emerging of new variants has accelerated after the Farmington strain in 2002 [7]. However, the mechanism for the evolution of NoV GII.4 variants and the factors involved in the evolution are still poorly understood.

It has been revealed that the histo-blood group antigens (HBGAs) serve as the recognition receptor for NoV entering into human host cell [21–23]. HBGAs are glycans composed of various carbohydrate moieties as well as different covalent linkages between the carbohydrates, which can be detected in human red blood cells, saliva, and epithelial cells [7, 24]. The synthesis of HBGAs proceeds through sequential addition of monosaccharides to the precursor disaccharide with the catalyzation of various glycosyltransferases, which results in numerous types of HBGAs. The α1,2-fucosyltransferase encoded by FUT2 gene catalyzes the formation of the H type HBGAs, which can be further catalyzed by enzyme A, enzyme B and Lewis α1,3-fucosyltransferase (FUT3) to form the A type, B type, Le$^b$and Le$^y$ phenotypes. All these types of HBGAs contain the α1,2 Fuc saccharide catalyzed by the α1,2-fucosyltransferase, and they are termed as secretor types [24]. For the individuals with defects in the FUT2 gene, the secretor HBGAs cannot be expressed but the Le$^a$ and Le$^x$ non-secretor HBGA types can be found in their body fluids or intestinal cells [7]. Many experimental works have found that GII.4 NoVs recognize and interact with these types of HBGAs in a strain specific manner. Different GII.4 strains have different preferences for these HBGA types with different binding affinities.

HBGAs have been believed to be one of the factors that direct the evolution of GII.4 NoVs [9, 25, 26], and therefore the investigation of the binding interactions between GII.4 variants and HBGAs is helpful for better understanding the evolution of GII.4 NoVs. The study of Lindesmith et al. found that the HBGA-binding profiles have been changed along with the evolution of GII.4 variants [9]. Whereas, the research of Yang et al. revealed that compared with other genogroups and genotypes of NoV, GII.4 strains can bind with a broad range of HBGAs and the binding ability persisted during the GII.4 NoV evolution [25]. The X-ray crystallographic investigations of Singh et al. showed that GII.4 NoV strains can recognize numerous types of HBGA and many of these binding patterns were not detected in the earlier binding assays [26]. In these previous experimental GII.4 NoV-HBGAs binding interaction studies, only the GII.4 variants before 2006b strain were investigated and somewhat conflicted conclusions were deduced. The results of the NoV-HBGAs binding assays are also not well consistent with the X-ray crystallographic observations [9, 25, 26]. Therefore, the evolution of the HBGA-binding affinities for the GII.4 NoV variants, especially including the recent occurred strains, should be further explored. In addition, it is believed that HBGAs play an important role in selecting the predominate variants during the evolution of GII.4 NoVs, however, the molecular mechanism behind the evolution of the NoV-HBGAs interactions is still not well understood. In this study, the evolution of the interactions between GII.4 NoV strains and HBGAs was measured by using HBGA-binding assays, and the molecular mechanism responsible for the evolution of the NoV-HBGAs interactions was investigated by using the molecular dynamics (MD) simulations combined with the molecular mechanics/generalized born surface area (MMGBSA) method.

In the present work, the virus-like particles (VLPs) of nine representative GII.4 strains circulated over the past decades, i.e. US 95/96, Farmington Hills, Hunter, 2006a, 2008, New Orleans 2009 (named as New Orleans in the remainder of the paper), Sydney 2012 (named as Sydney), Sydney 2012 Hong Kong (abbreviated as Hong Kong) and Sydney 2012 GII.P16/

GII.4 (abbreviated as GII.P16/GII.4), were produced by using the *Hansenula polymorpha yeast* expression platform constructed by our laboratory, and the relative binding affinities for these GII.4 variants with different types of HBGA were determined by the enzyme linked immuno-sorbent assay (ELISA)-based HBGA-binding assays. In order to improve the reliability of the measurements and detect the low-affinity interactions, the binding assays were performed repeatedly under several different concentrations of GII.4 VLPs. Based on the experimental results, the changes in HBGA-binding affinities along with the evolution of GII.4 variants were analyzed. Furthermore, in order to investigate the mechanism determining the changes of HBGA-binding affinities for these GII.4 NoV strains, the MD simulations and MMGBSA calculations were carried out. Our computational studies indicated that several key residue mutations and the associated conformational adjustments account for the changes in HBGA-binding affinities during the evolution of GII.4 NoVs.

## Results

### Production and morphological observation of the VLPs for the various representative GII.4 NoV strains

The VP1 proteins of nine representative GII.4 NoV strains that were emerged and prevalent in different periods, including US 95/96, Farmington Hills, Hunter, 2006a, 2008, New Orleans, Sydney, Hong Kong and GII.P16/GII.4, were produced by using the *Hansenula polymorpha* yeast expression platform developed by our laboratory [27–29]. The expressed VP1 proteins were obtained through purification with the ion exchange chromatography method. In order to confirm the self-assembly of the VP1 proteins into VLPs, the purified samples were observed with the negative-stained transmission electron microscope (TEM). It is found that the VP1 proteins of all these nine strains are successfully self-assembled into particles with uniform morphology, as shown in Fig 1. These results demonstrate that all of the VP1 proteins of these nine GII.4 NoV strains can be effectively expressed with high yields and also self-assembled into VLPs by using our *Hansenula polymorpha* yeast expression platform.

### The evolution of HBGA-binding affinity for different GII.4 NoV epidemic strains measured by ELISA-based binding assays

In our VLP-HBGA binding assays, the biotinylated polyacrylamide-conjugated carbohydrates (HBGA-PAA-biotin) were coated onto the well walls of the plate, and series of diluted VLP samples were added into the HBGA-coated wells to evaluate the binding of VLPs with the HBGAs under different VLP concentrations. In the present study, a total of nine types of HBGA and nine GII.4 NoV strains were used in the binding assays to detect the different interactions between them. The experimental results are shown in Fig 2. It is found that different NoV strains exhibit various binding strengths to different HBGA types. For type A HBGA, the US 95/96, Farmington Hills and Hunter strains, which are epidemic in the early years, exhibit lower binding activities. Whereas the newly emerged strains, i.e. New Orleans, Sydney, Hong Kong and GII.P16/GII.4, have the strongest binding affinities to type A HBGA. The 2006a strain also exhibits strong binding activity, and the 2008 strain displays a moderate binding affinity, as shown in Fig 2A. For type B HBGA, the older strains, including US 95/96, Farmington Hills, Hunter strains and 2008 strains, have relative lower binding activities, whereas the relative newly epidemic strains, containing 2006a, New Orleans, Sydney, Hong Kong and GII.P16/GII.4, have relative stronger interactions with the receptor, as displayed in Fig 2B. Comparing Fig 2A and 2B, it is found that the binding of type A HBGA to the favorable GII.4 NoV strains is generally stronger than that of type B, where the binding curves for type A almost

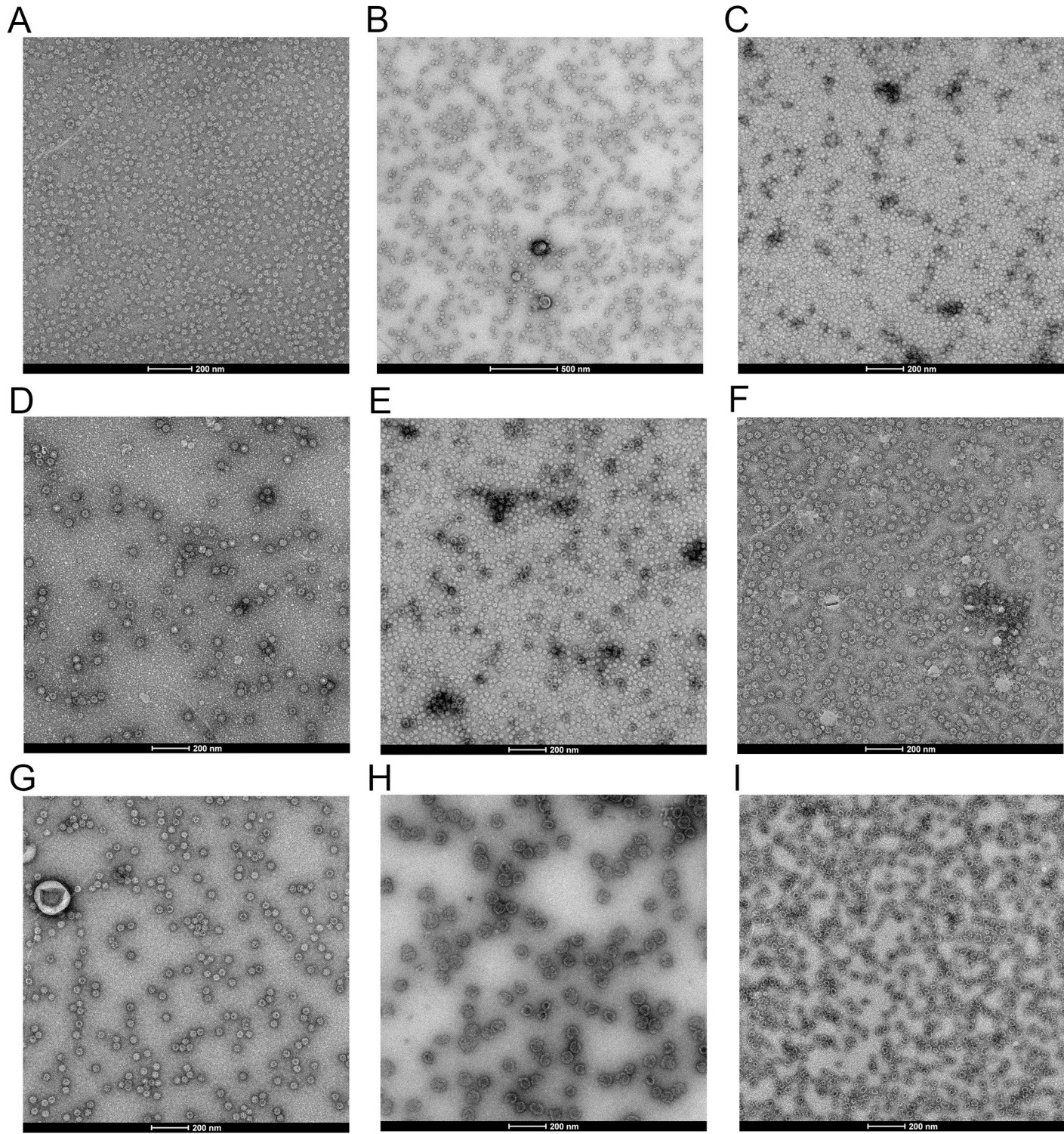

**Fig 1. The morphological observation of the VLPs recombinantly expressed by the *Hansenula polymorpha* yeast for the nine representative GII.4 NoV strains.** (A-I) display the morphology observed by TEM for the US 95/96, Farmington Hills, Hunter, 2006a, 2008, New Orleans, Sydney, Hong Kong and GII.P16/GII.4 strains, respectively.

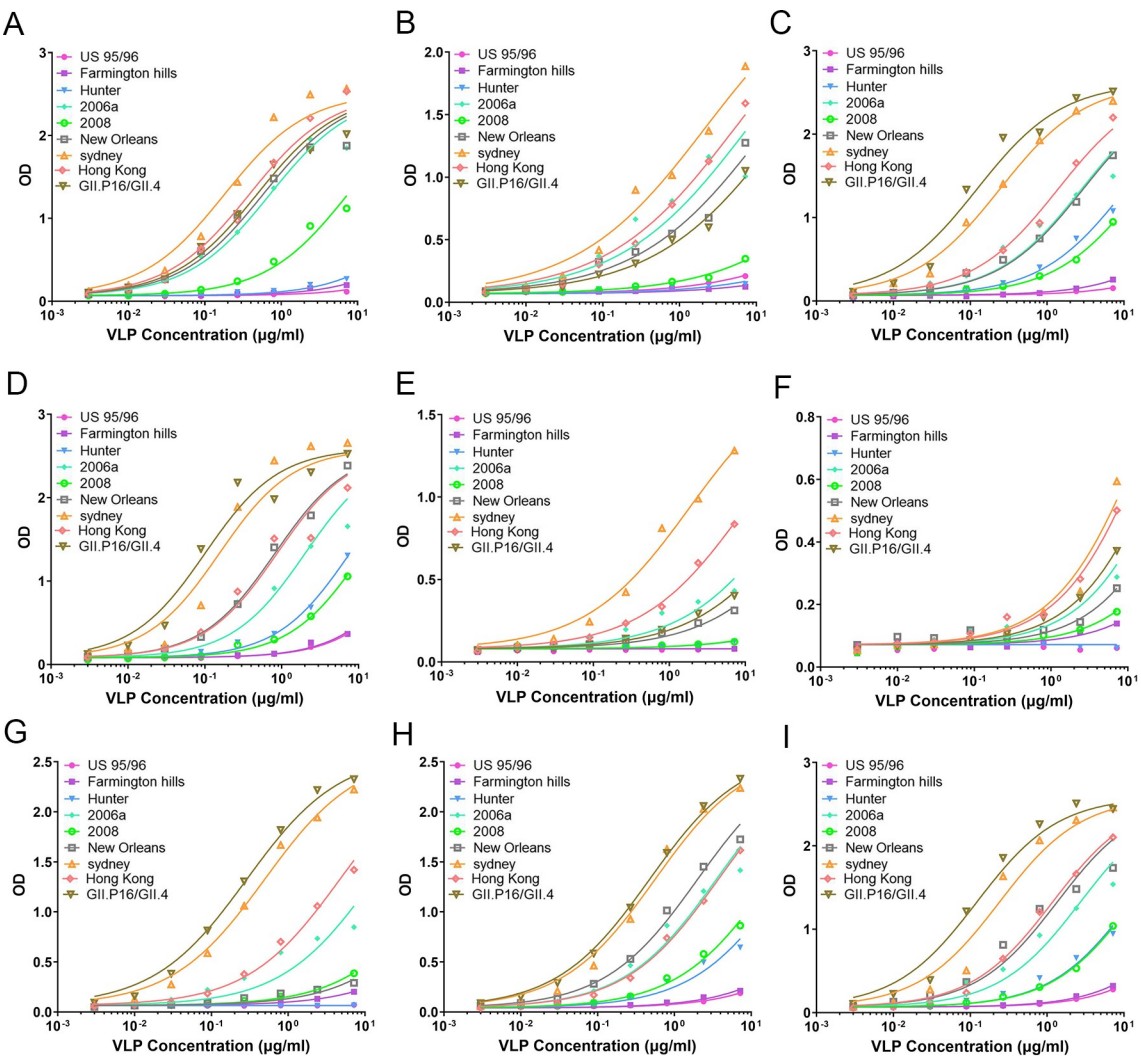

**Fig 2. The binding of GII.4 NoV VLPs to HBGAs measured by ELISA-based binding assays.** (A-I) display the binding ability of the VLPs of the nine GII.4 strains to the HBGA types A, B, H1, H3, H2, Le$^a$, Le$^x$, Le$^y$, Le$^b$, respectively. In these figures, the x axis denotes the different concentration of the VLP samples and the y axis indicates the HBGA binding ability.

reach saturation but those of type B are far from reaching the saturation point even at a very high concentration of NoV VLP samples. The HBGA types H1 and H3 exhibit the similar binding profiles to the GII.4 NoV variants as displayed in Fig 2C and 2D, where the US 95/96 and Farmington Hills strains have very low binding activities, the Hunter and 2008 strains display moderate binding intensities, and the other relative new strains have strong binding affinities. For the HBGA types H2 and Le$^a$, the US 95/96 and Hunter strains display no-binding activities, whereas Sydney and Hong Kong strains exhibit relative higher binding strengths (See Fig 2E and 2F). Compared with other HBGA types, the H2 and Le$^a$ exhibit relative lower binding intensities to GII.4 NoV strains, where the binding curves do not reach the saturation point even at the very high concentration of the samples, as shown in Fig 2E and 2F. For the HBGA type Le$^x$, the US 95/96 and Hunter strains have no-binding activities, Farmington Hills, 2008 and New Orleans strains exhibit low binding activities, 2006a and Hong Kong strains display moderate binding intensities, and Sydney and GII.P16/GII.4 strains have relative higher binding affinities as shown in Fig 2G.

All these above HBGA types belong to disaccharide or trisaccharide. Besides these disaccharides and trisaccharides, we also performed the binding assays for two tetrasaccharides including Le$^y$ and Le$^b$ types, as shown in Fig 2H and 2I. The binding profiles to the GII.4 NoV variants of these two HBGA types are very similar. The binding activities for the US 95/96 and Farmington Hills, which are the oldest epidemic GII.4 strains, are low. The binding intensities for Hunter and 2008 strains are moderate, while the binding affinities for the other strains are relative stronger. Especially, the newly epidemic Sydney and GII.P16/GII.4 strains exhibit the highest binding affinities to the Le$^y$ and Le$^b$ types HBGA.

In summary, the VLP-HBGA binding assays indicate that for all the disaccharide, trisaccharide and tetrasaccharide HBGAs, the early epidemic strains, such as the US 95/96 and Farmington Hills strains, have low binding activities, whereas the newly emerged strains including Sydney, Hong Kong and GII.P16/GII.4 strains exhibit relative stronger binding intensities. Our experimental results suggest that the evolution of GII.4 NoVs results in the increased binding affinity as well as the broader binding profile to the receptor HBGAs, which demonstrates that HBGAs may play important roles in selecting the predominate variants in the evolution of GII.4 NoVs.

## The evolution of the GII.4 NoV-HBGA binding affinity revealed by MD simulations combined with MMGBSA method

In order to investigate the molecular mechanism responsible for the discrepancies in the HBGA-binding affinity for different GII.4 NoV strains, the atomic interactions between the corresponding NoV capsid VP1 proteins and the receptor HBGAs were analyzed in detail by using MD simulations and MMGBSA calculations [30, 31]. In our studies, five representative GII.4 NoV strains were selected to be analyzed including two early epidemic strains (i.e. the US 95/96 and Farmington Hills strains), which exhibit relatively low-binding activities to HBGAs, and three newly emerged strains (i.e. the Sydney, Hong Kong and GII.P16/GII.4 strains) that have the strongest binding affinities to HBGAs. For HBGAs, two types, i.e. types A and Le$^y$, were studied in the present work, in which type A is a trisaccharide and type Le$^y$ is a tetrasaccharide. The NoV capsid is assembled by 180 copies of VP1 proteins, which form an icosahedral shell. Each VP1 protein is composed of two domains: P domain and S domain. The S domain is responsible for the assembly of the capsid, which is located in the interior of the shell, and the P domain protruding out of the shell is responsible for the binding of receptor [32, 33]. X-ray crystallographic studies have revealed that the HBGA receptor binding site is located at the dimeric interface of two adjacent P domains [34]. The HBGA binding pocket is composed of loop-1 (residues 372–378), loop-2 (residues 390–396), loop-3 (residues 441–445) and an β-sheet (residues 341–346), in which loop-2 and loop-3 belong to one monomer, and loop-1 and the β-sheet are from the adjacent monomer, as shown in Fig 3A. Sequence alignments between the VP1 proteins of the five studied strains show that the residue divergences between these strains are mainly located at the P domain. Especially, several mutated residues are around the HBGA-binding pocket, as shown in Fig 3B.

In this study, based on the crystallographically resolved complex structures of Saga4 strain with PDB codes 4x07 and 4wze [26], the P domain-HBGA complex structures for the studied representative GII.4 NoV strains were built by using the UCSF chimera software [35], as described in the "Materials and Methods" section. These constructed complex structures were used as the initial structures for MD simulations, which are displayed in Fig 4A. Then, 20ns MD simulation was performed for each complex system. To examine the stability of the simulations, the time-evolution of the root mean square deviation (RMSD) of all atoms in the protein complex was calculated (see S1 Fig in the Supporting information). The RMSD values

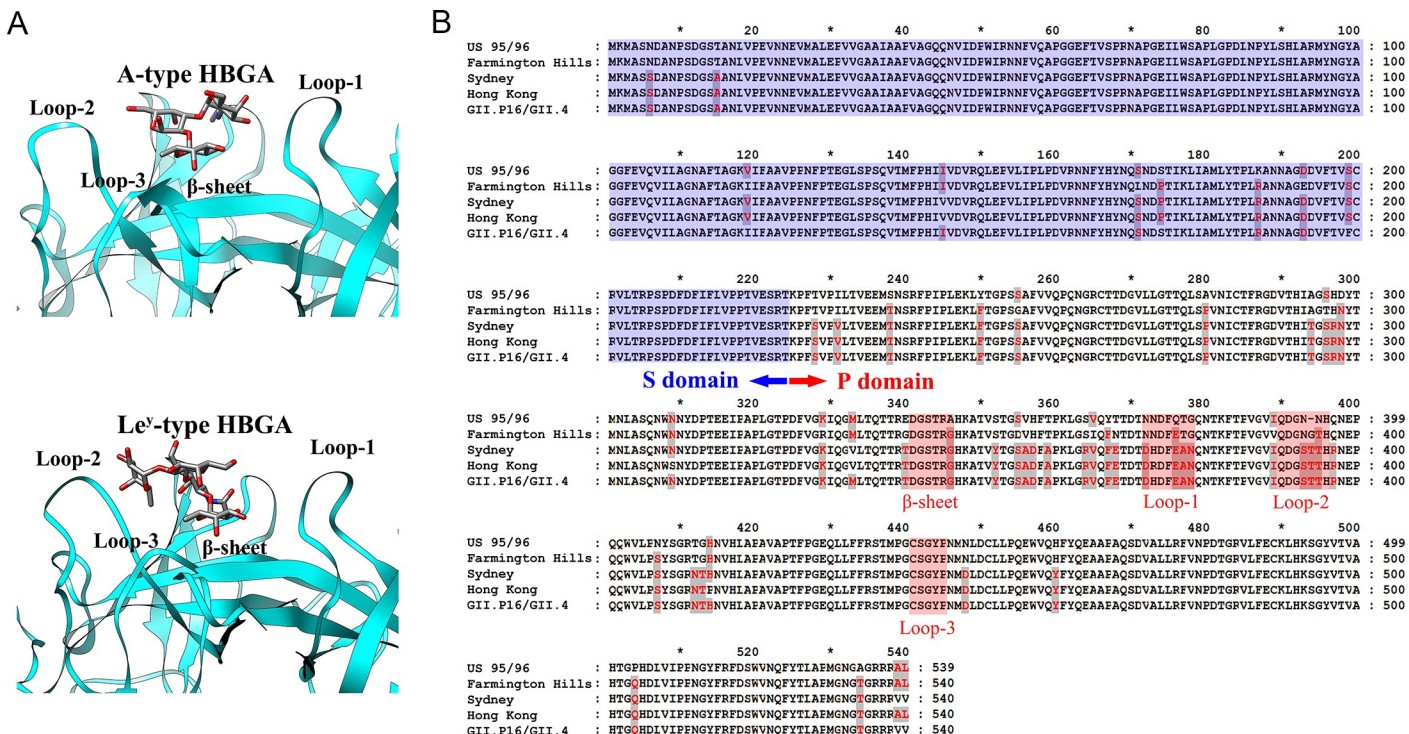

**Fig 3. The sequences and tertiary structure of the HBGA binding pocket.** (A) The tertiary structures of the dimeric P domains of GII.4 NoV in complex with type A and type Le$^y$ HBGAs, respectively, in which the HBGA binding pocket is formed by loop-1 (residues 372–378), loop-2 (residues 390–396), loop-3 (residues 441–445) and an β-sheet (residues 341–346) marked in the structure. The structures are visualized by using the UCSF chimera software (developed by the Resource for Biocomputing, Visualization, and Informatics at the University of California, San Francisco, with support from NIH P41-GM103311.) based on the coordinates of the GII.4 NoV Saga4 strain complexed with type A (PDB code: 4x07) and type Le$^y$ (PDB code: 4wze) HBGAs, respectively. (B) Sequence alignments between the VP1 proteins of the five studied GII.4 strains. The S and P domains of the VP1 proteins are marked, and the residues involved in the receptor binding pocket, i.e. loop-1, loop-2, loop-3 and an β-sheet, are shaded by pink color in the figure.

were computed with respect to the starting structure in the production MD simulations. The RMSD plots show that all the simulated complex structures were quite stable throughout the simulations. Based on the simulation trajectories, the binding free energy between HBGAs and the P domains of various GII.4 strains were calculated with MMGBSA method. The calculation results are displayed in Fig 4. For the type A trisaccharide HBGA, the oldest US 95/96 strain has the weakest binding affinity, where the binding free energy is -29.07 kcal/mol, and the Farmington Hills strain exhibits a relative stronger binding strength with the free energy of -31.79 kcal/mol as shown in Fig 4B. Whereas the newly epidemic strains, i.e. the Sydney, Hong Kong and GII.P16/GII.4, have the strongest binding affinities with type A HBGA. It is also found that the differences in the binding free energy between these three newly emerged strains are negligible, where the values of binding free energy are -34.90 kcal/mol, -34.80 kcal/mol and -34.65 kcal/mol, respectively. The simulation results are well consistent with our experimental data as discussed above. Both the experimental and simulation results indicate that along with the evolution of GII.4 NoVs, the binding affinity with type A HBGA is distinctly increased. The similar results are also obtained for the type Le$^y$ tetrasaccharide HBGA. The lower sub-figure in Fig 4B displays the simulation results for the binding of Le$^y$ HBGA. The early epidemic US 95/96 and Farmington Hills strains have the relative lower binding activities with the receptor Le$^y$ HBGA, and the binding strength of the Farmington Hills strain is slightly greater than that of the US 95/96 strain, in which the binding free energies for these two strains are -36.17 kcal/mol and -36.41 kcal/mol, respectively. However, the binding affinity

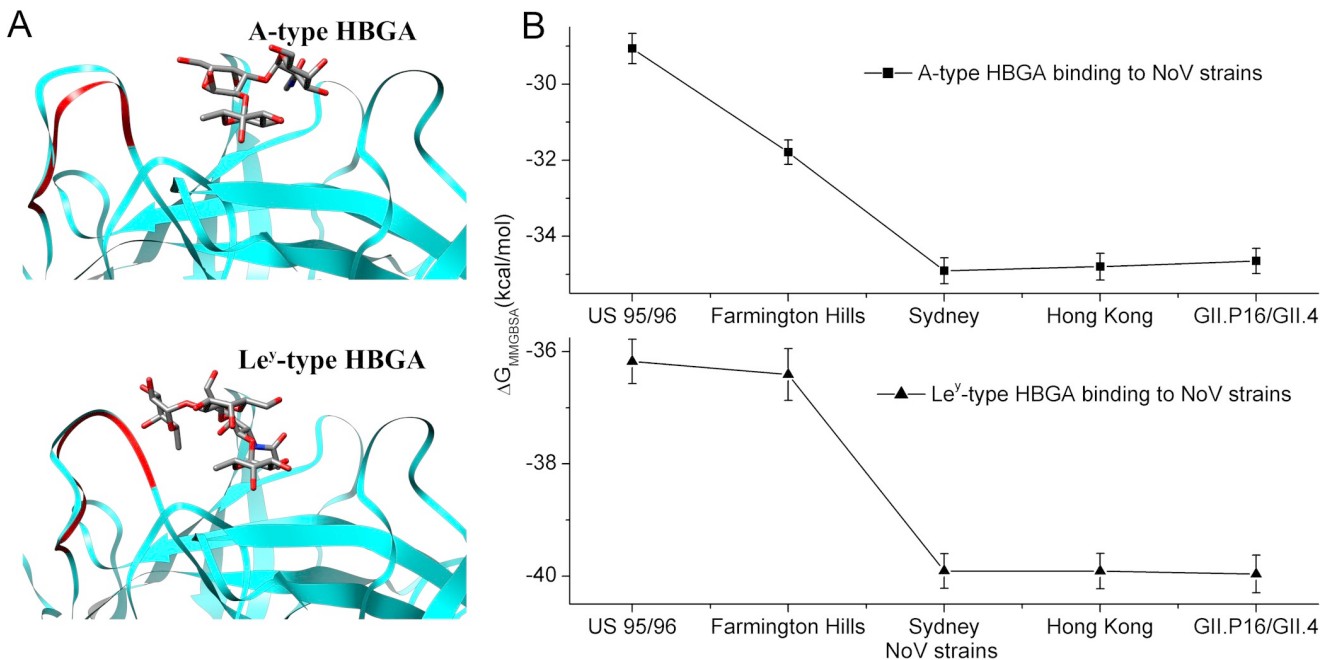

**Fig 4. The evolution of HBGA binding affinity for the P domains of the various GII.4 NoVs revealed by MD simulations and MMGBSA calculations.** (A) The initial structures of the P domains complexed with type A (upper sub-figure) and type Le$^y$ HBGAs (lower sub-figure) in the MD simulations for the five studied GII.4 strains, i.e. US 95/96, Farmington Hills, Sydney, Hong Kong and GII.P16/GII.4. These complex structures were constructed based on the same templates as described in the main text. Therefore, the main chain of the structure is the same for all these studied strains expect for the US 95/96 strain. Owing to the lack of a residue (residue No. 394) in loop-2 for the US 95.96 strain, the length of loop-2 in this strain (displayed in red) is shorter than that of the other strains. (B) The binding free energy calculated by MD simulations combined with MMGBSA method for the various GII.4 strains interacted with type A (upper sub-figure) and type Le$^y$ (lower sub-figure) HBGAs, respectively. Data are presented as the mean value along with the standard error of the mean (SEM) calculated from 100 snapshots of the MD simulation for each system.

is significantly improved for the three newly emerged strains, i.e., the Sydney, Hong Kong and GII.P16/GII.4 strains, where the binding free energies are -39.90 kcal/mol, -39.91 kcal/mol and -39.96 kcal/mol, respectively. In addition, the binding free energy differences between these three strains are very small. The simulation results agree well with our experimental results obtained by binding assays. These above experimental and simulation results demonstrate that both for the trisaccharide type A and the tetrasaccharide type Le$^y$ HBGAs, NoV evolutions result in a distinctly increased binding affinity to the receptors, implying that HBGAs may play important roles in the evolution of GII.4 NoVs. Our study results agree with the observations in the surface plasmon resonance experiments carried out by de Rougemont et al. [7]. They revealed that the more recent post-2002 GII.4 NoV variants exhibit a stronger affinity to HBGAs than the pre-2002 strains [5, 7].

### The molecular mechanism for the evolution-induced improvements of the NoV-HBGAs binding affinity investigated by MD simulations and MMGBSA calculations

In order to study the molecular mechanism for the improvements of the NoV-HBGAs binding strength, the per-residue energy decomposition analysis was performed to reveal which residues mainly contribute to the increased binding affinity between the NoV P domains and HBGAs, and the calculation results are shown in Fig 5. It is found that both for type A and type Le$^y$ HBGAs, the residues Ser343, Thr344 and Arg345 of the β-sheet in the binding pocket exhibit the biggest contributions to the NoV-HBGA binding affinity. However, these residues

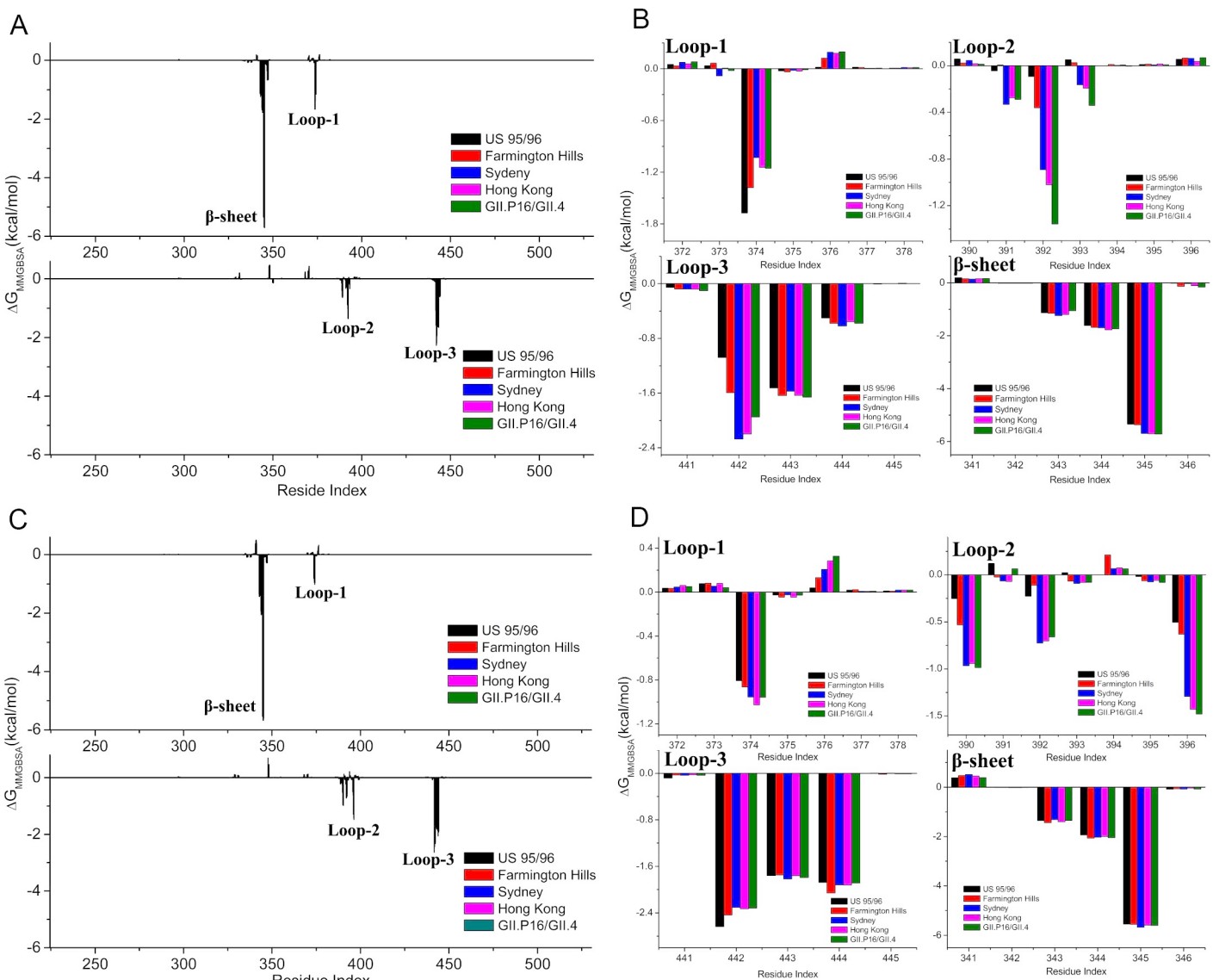

**Fig 5. The per-residue energy decomposition analysis results.** (A) The per-residue contribution to the binding of type A HBGA for the various GII.4 strains. The upper and lower sub-figures correspond to the two dimeric P domains. (B) The type A HBGA binding free energy contributed by the residues in the binding pocket including loop-1 (residues 372–378), loop-2 (residues 390–396), loop-3 (residues 441–445) and an β-sheet (residues 341–346). (C) The per-residue contribution to the binding of type Le^y HBGA for the various GII.4 strains. The upper and lower sub-figures correspond to the two dimeric P domains. (D) The type Le^y HBGA binding free energy contributed by the residues in the binding pocket, i.e. loop-1 (residues 372–378), loop-2 (residues 390–396), loop-3 (residues 441–445) and an β-sheet (residues 341–346).

are conserved and their interactions with HBGAs almost have no changes for the various GII.4 NoV strains as shown in Fig 5, and therefore the residues of this β-sheet are not responsible for the increasing binding affinity during the evolution of GII.4 strains. The residues in loop-3 including Ser442, Gly443 and Tyr444 also have significant interactions with the receptors, and these interactions with the type Le^y HBGA also have negligible changes during the evolution of the GII.4 strains as shown in Fig 5. But for the type A HBGA shown in Fig 5A and 5B, the binding interaction strength contributed by the residue Ser442 in loop-3 is improved along with the evolution of GII.4 strains, in which the binding affinity of the Farmington Hills strain is higher than that of the US95/96 strain, and the interaction strength is further increased for

the new strains including Sydney, Hong Kong and GII.P16/GII.4. The residue Asp374 in loop-1 also play important roles in the recognition both of the types A and Le$^y$ HBGAs, and the interaction strength has little difference between the various GII.4 strains, as shown in Fig 5B and 5D. From Fig 5, it is found that the increased NoV-HBGAs binding affinity during the evolution of GII.4 strains is predominantly caused by the interactions involving loop-2. Exactly, for the type A HBGA the receptor-binding interactions of the residues Asp391, Gly392 and Asn393 (or Ser393) are significantly improved, and for the type Le$^y$ HBGA the increased receptor-binding affinity mainly attributes to the residues Gln390, Gly392 and His396. An interesting and important point should be noted that except for the residue 393, all these residues directly interacted with the receptors are conserved across the studied GII.4 NoV variants, which may be important for the binding specificity and affinity of NoVs to the receptor HBGAs. Thus, there must exist other residues that are not directly interacted with the receptor, whose mutations indirectly improve the HBGA-binding affinity through increasing the interactions between these directly interacted residues and the receptors. De Rougemont et al. [7] and Lindesmith et al. [9] also found that the residues that are not directly involved in the HBGAs attachment distinctly contribute to the binding affinity of GII.4 NoVs to the receptor HBGAs.

Sequence alignments between the various GII.4 NoV strains have shown that a total of three residue mutations occurred in loop-2 during the evolution of the GII.4 NoVs, including the residues 393, 394 and 395. From the US95/96 strain to the Farmington Hills strain, a new residue Gly394 was inserted into loop-2 and the residue Asn395 was mutated to Thr. From the Farmington Hills strain to the Sydney strain, the inserted residue Gly394 was further mutated to Thr as well as the residue 393 was mutated from Asn to Ser. For the Sydney, Hong Kong and GII.P16/GII.4 strains, no further residue mutation occurred between them. De Rougemont et al. also revealed that the changes of the residue 395 that is not directly interacted with HBGA can distinctly regulate the binding properties of GII.4 NoVs [7]. Then, based on the MD simulation trajectories, we investigated why the mutations of these residues, which are not directly interacted with the receptor, result in the increased HBGA-binding affinity. The average NoV P domain-HBGA complex structures in the MD simulations for the various studied GII.4 strains were superposed, and the conformations of loop-2 for these different strains were compared in Figs 6 and 7. For the structures in complex of the type A HBGA, loop-2 is closer to the receptor in the Farmington Hills strain compared with the US 95/96 strain, which is further significantly closer to the HBGA from the Farmington Hills strain to the Sydney, Hong Kong and GII.P16/GII.4 strains, as shown in Fig 6. The conformational rearrangements of loop-2, especially the residues 391, 392 and 393, relative to the receptor are responsible for the increasing NoV-HBGA binding affinity during the evolution of the GII.4 NoV strains. For the US95/96 strain, the residue contacts in loop-2 were analyzed by using the "Find Clashes/Contacts" tool of the UCSF Chimera software [35], in which the cutoff and allowance values are set to 0.4 Å and 0.0 Å, respectively. It is found that the residues Gly392 and Asn395 form contacts with the residues Gln390 and Asn398, and these residue interactions result in the top of loop-2 swinging out of the receptor-binding pocket, as shown in Fig 6B. Especially, the part of loop-2 nearest to the receptor, including the residues Asp391, Gly392 and Asn393, moves away from the receptor, which is responsible for the lowest binding affinity to HBGA. From the US95/96 strain to the Farmington Hills strain, a new residue Gly394 was inserted into loop-2, which make the loop longer. The contacts formed by the residues Gln390, Asn393, Thr395 and Asn398 also cause the top of loop-2 swing out of the binding pocket, as shown in Fig 6C. However, the swing of the longer loop-2 in the Farmington Hills strain does not draw the part of loop-2 near the receptor as far away from the receptor as that of the US95/96 strain. Specifically, the residue Gly392 is closer to the receptor in the Farmington Hills strain than

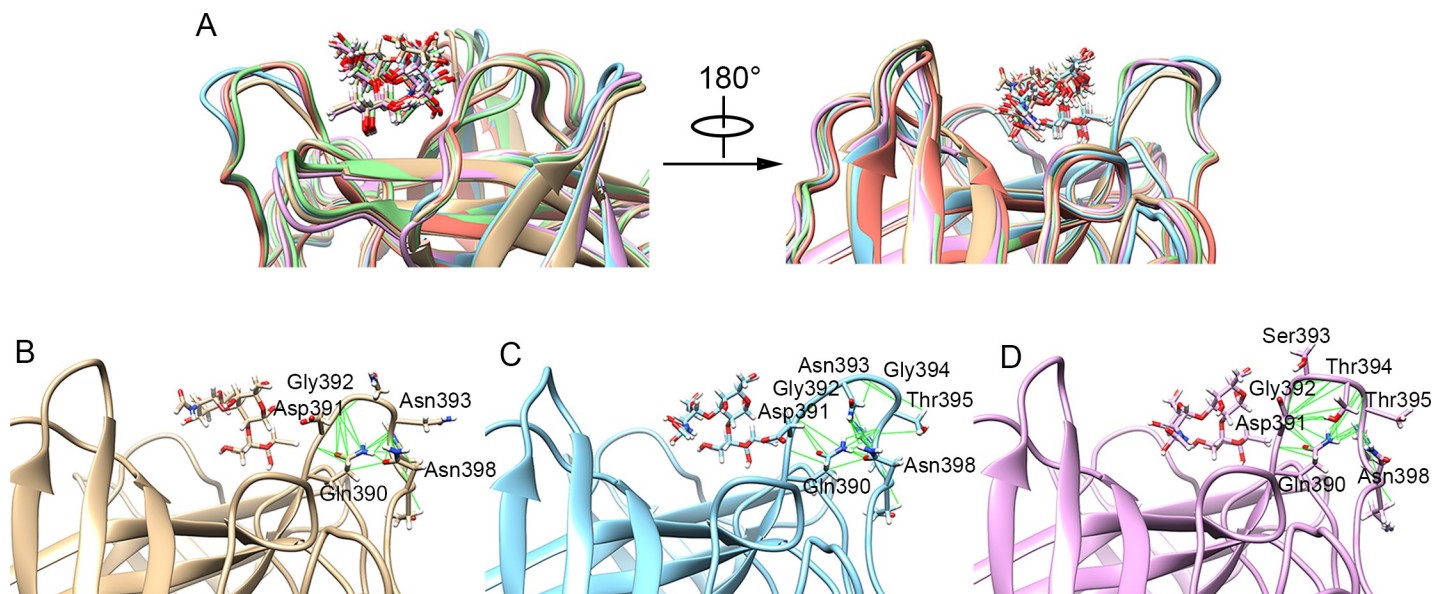

**Fig 6. The conformational rearrangements of loop-2 for the various GII.4 strains binding to the receptor type A HBGA.** (A) The comparison of the conformations of loop-2 in the average P domain-HBGA complex structure of the MD simulations for the various studied GII.4 strains. The average P domain-HBGA complex structures of the MD simulations for the various strains were superposed by using the UCSF chimera software. The structures of the US95/96, Farmington Hills, Sydney, Hong Kong and GII.P16/GII.4 strains were displayed in gray, blue, purple, green and pink colors, respectively. (B-D) show the residue contacts in loop-2 analyzed by using the "Find Clashes/Contacts" tool of the UCSF Chimera software for the US95/96, Farmington Hills and Sydney strains, respectively.

that of the US95/96 strain, which accounts for the improved receptor-binding affinity for the Farmington Hills strain. For the Sydney strain compared with the Farmington Hills strain, the inserted residue Gly394 is mutated to Thr and the residue Asn393 is mutated to Ser. In the residue interaction network of loop-2, the position occupied by the sidechain of Asn393 is replaced by the Gly394 sidechain in the Sydney strain compared with that in the Farmington Hills strain. Besides that, the residue 393 is mutated from Asn to Ser, whose sidechain is transformed from the conformation pointing out of the binding pocket into the inward conformation. These above conformational changes result in the receptor-interacting part of loop-2, especially the residues Asp391, Gly392 and Ser393, significantly closer to the receptor HBGA, which is responsible for the distinctly increased NoV-HBGA binding affinity for the Sydney strain. The loop-2 conformations and the intra-loop interactions for the Hong Kong and GII. P16/GII.4 strains are similar to those of the Sydney strain (see S2 Fig in the Supporting information), which accounts for the similar receptor binding affinities for these three strains.

For the structures in complex of the type Le$^y$ HBGA as shown in Fig 7, the part of loop-2 near the receptor composed of the residues 390–394 is significantly closer to the receptor in the Farmington Hills strain than that in the US 95/96 strain, which is further closer to the receptor in the three Sydney related strains (i.e. Sydney, Hong Kong, and GII.P16/GII.4 strains). Similar to the complex systems with the type A HBGA, the conformational rearrangements in loop-2 are also responsible for the increasing binding affinity in the type Le$^y$ HBGA-complexed structures during the evolution of the GII.4 NoV strains. In the US 95/96 strain, the interactions formed by the residues Asn393 and Asn395 with the residues Gln390 and Asn398 cause the loop-2 tilting outside the binding pocket, which result in the lowest binding strength with the receptor Le$^y$ HBGA. Our simulation results are well consistent with the X-ray crystallographic evidences that the loop-2 in the older 1996 strain complexed with HBGA is positioned out of the receptor binding pocket compared with the more recent 2004 strain,

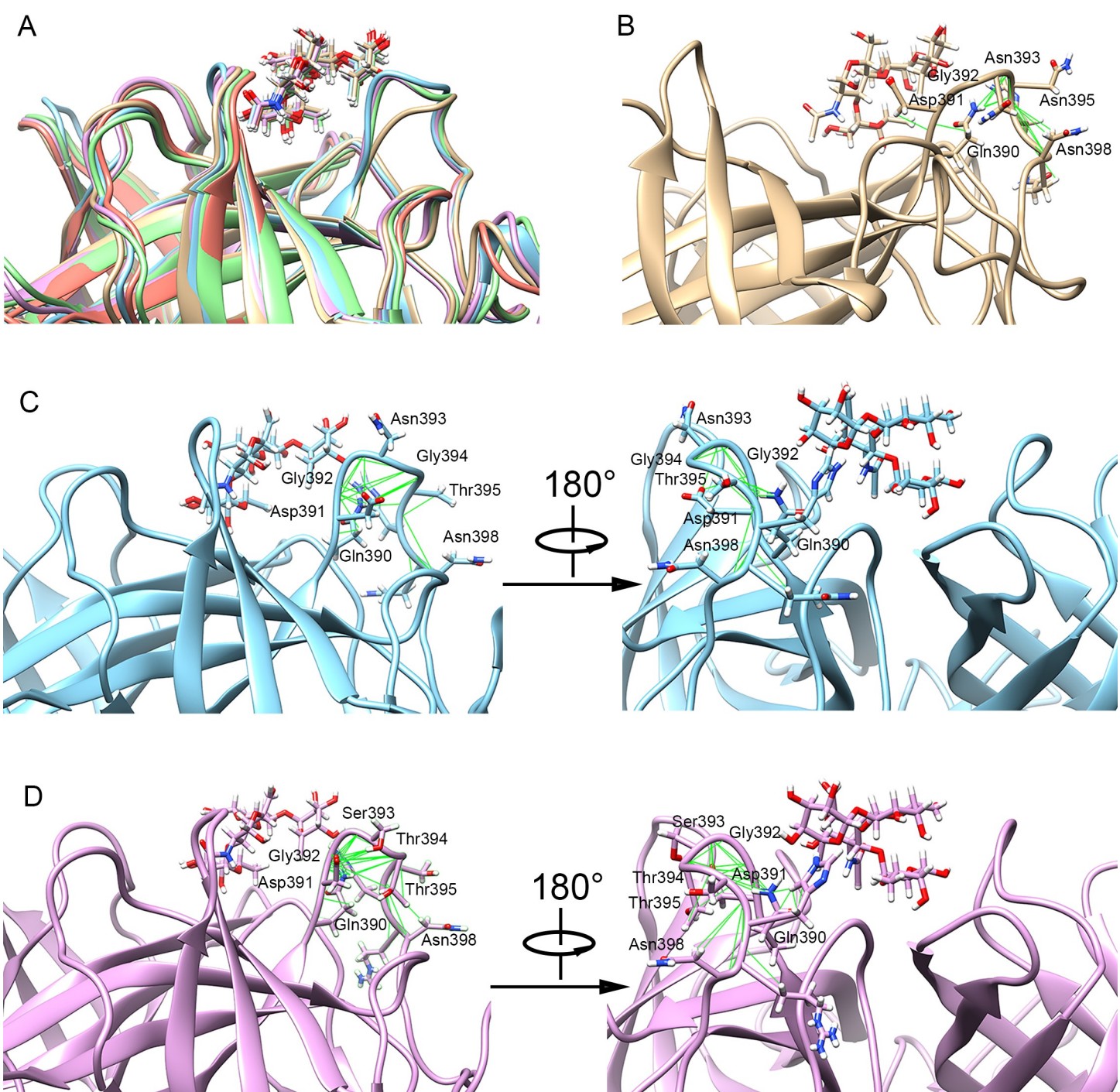

**Fig 7. The conformational rearrangements of loop-2 for the various GII.4 strains binding to the receptor type Le$^y$ HBGA.** (A) The comparison of the conformations of loop-2 in the average P domain-HBGA complex structure of the MD simulations for the various studied GII.4 strains. The average P domain-HBGA complex structures of the MD simulations for the various strains were superposed by using the UCSF chimera software. The structures of the US95/96, Farmington Hills, Sydney, Hong Kong and GII.P16/GII.4 strains were displayed in gray, blue, purple, green and pink colors, respectively. (B-D) show the residue contacts in loop-2 analyzed by using the "Find Clashes/Contacts" tool of the UCSF Chimera software for the US95/96, Farmington Hills and Sydney strains, respectively.

which results in the significantly reduced HBGA binding affinity for the 1996 strain [5]. In the Farmington Hills stain, the intra-loop interactions only induce the bending of the outer-side part of loop-2, and the part near the receptor does not bend together away from the pocket owing to the longer length of the loop, as displayed in Fig 7B. In the Sydney strain, the conformation of loop-2 undergoes a small twisting movement compared with the Farmington Hills strain, which results in the outer-side of loop-2 being further apart from the binding pocket and the part of loop-2 near the receptor moving closer to HBGA, as shown in Fig 7C. Especially, the residue Gly392 is obviously closer to HBGA in the Sydney strain than that in the Farmington Hills strain, which is responsible for the distinct improvement of the binding strength contributed by the residue Gly392 in the Sydney strain displayed in Fig 5. Besides that, the twisting motion of loop-2 also induces the sidechains of the residues Gln390 and His396 moving closer and forming atomic contacts between them. The conformational rearrangements of the residues Gln390 and His396 result in these two residues being closer to the receptor HBGA and forming new hydrogen bonds between His396 and the HBGA. The occupancy rate of two hydrogen bonds between Gln396 and HBGA is increased from 11.0% and 2.4% to 23.1% and 11.1%, respectively, in the MD simulation trajectory. The approximation of these residues to HBGA as well as the formation of the hydrogen bonds account for the significantly improvement of the Le$^y$ HBGA-binding affinity for the Sydney NoV strain in comparison with the Farmington Hills strain. The conformation of loop-2 in the Hong Kong and GII. P16/GII.4 strains is almost the same as that in the Sydney strain (see S3 Fig in the Supporting information), which provides the structural mechanism for the similar HBGA-binding affinities of these three NoV strains.

Taken together, the above results indicate that during the evolution of GII.4 NoV strains, the residues directly interacted with the receptor HBGAs were conserved, which may be important for the specificity and affinity in the receptor recognition. The increasing HBGAs-binding affinity is mainly caused by the mutations of the residues without direct interactions with HBGAs. Based on the MMGBSA calculations and the MD simulation trajectories analyses, a novel molecular mechanism was proposed for the evolutionary increasing of the HBGA-binding affinity. The evolution-involved residue mutations trigger the conformational rearrangements of loop-2, which results in the narrowing of the pocket and thus the tightening of the receptor binding. Our simulation results are consistent with the study of Lindesmith et al. [9], which proposed that the HBGA binding affinity and specificity are likely governed by the subtle sequence and conformational changes in loop-2. They also pointed out that the mutation of the residues which are not directly involved in the binding of the receptor can drastically alter the HBGA binding affinity through changing the interactions with the receptor binding residues. De Rougemont et al. also found that the changes of the residues that are not directly interacted with HBGA can obviously modulate the receptor binding affinity and specificity [7].

## Discussion

NoV is one of the leading causes for the outbreaks of viral gastroenteritis in humans around the world, which threatens the health both of children and adults. NoVs are highly genetically diverse, and new genotypes as well as even new genogroups of NoV have been continuously emerged, among which the GII.4 genotype is the most predominant over the past decades. GII.4 genotype also evolve rapidly and on average every 2–3 years a new strain will emerge to replace the previously prevalent variant. It has been revealed that HBGAs serve as the recognition receptor for the GII.4 NoVs infecting the host cell, and the interactions with HBGAs may play an important role in selecting the predominate variants by the nature in the evolution of GII.4 NoVs. In order to reveal the evolution of the HBGAs-binding affinity for the GII.4 NoV

variants and investigate the molecular mechanism behind the evolution of the NoV-HBGAs interactions, ELISA-based NoV VLP-HBGA binding assays as well as MD simulations combined with MMGBSA calculations were performed in the present work to study the interactions between various GII.4 strains and different types of HBGAs.

In our experimental studies, nine representative GII.4 NoV strains prevalent in the past decades, i.e. US 95/96, Farmington Hills, Hunter, 2006a, 2008, New Orleans, Sydney, Hong Kong and GII.P16/GII.4, and nine types of HBGAs, i.e. H type 3, type A (tri), type B (tri), Le$^d$(H type 1), H type 2, Le$^a$ and Le$^x$, Le$^b$ and Le$^y$, were selected to measure the relative binding affinities between them. Based on the experimental results of the HBGA-binding assays, the changes in HBGAs binding affinities along with the evolution of GII.4 variants were analyzed. It is found that for all the types of HBGAs, the evolution of GII.4 NoVs results in the increased binding affinity as well as the broader binding profile to these receptors. The early epidemic strains, such as the US 95/96 and Farmington Hills strains, have the lower binding activities, whereas the newly epidemic strains, i.e. Sydney, Hong Kong and GII.P16/GII.4 strains, exhibit relative stronger binding intensities. Our experimental results suggest that HBGAs may play important roles in the evolution of GII.4 NoVs.

In order to investigate the molecular mechanism responsible for the evolution of the NoV-HBGA binding affinity, the MD simulations and MMGBSA calculations were performed for five representative GII.4 NoV strains in complex with two types of HBGAs including the disaccharide A-type and the trisaccharide Le$^y$-type HBGAs. The MMGBSA calculation results show that both for type A and type Le$^y$ HBGAs, the oldest US 95/96 strain has the weakest binding affinity, and the Farmington Hills strain exhibits a relative stronger binding strength. Whereas, the binding affinity is significantly improved for the three newly emerged strains, i.e., the Sydney, Hong Kong and GII.P16/GII.4 strains. The calculation results are well consistent with our experimental data. The per-residue energy decomposition analyses show that the increased NoV-HBGAs binding affinity during the evolution of GII.4 strains is predominantly contributed by the interactions of loop-2. However, all these residues directly interacted with the receptors in loop-2 are conserved across the studied GII.4 NoV variants. The increasing HBGAs-binding affinity is mainly caused by the mutations of the residues without direct interactions with HBGAs, instead of the directly interacted residues. Based on the detailed analyses of the MD simulation trajectories, it is found that the evolution-involved residue mutations, which are not directly interacted with the receptor, trigger the conformational rearrangements of loop-2. These conformational rearrangements result in the narrowing of the pocket and thus the tightening of the receptor binding, which is responsible for the evolutionary increasing of the HBGA-binding affinity. Our studies provide the molecular mechanism for the increased HBGA-binding affinity during the evolution of GII.4 NoVs, which may be helpful for the drug and vaccine designs targeted to GII.4 NoVs.

## Materials and methods

### Production and morphological observation of the VLPs of the GII.4 NoV variants

The VP1 sequences of the nine representative GII.4 strains circulated in the past decades, i.e. US 95/96, Farmington Hills, Hunter, 2006a, 2008, New Orleans, Sydney, Hong Kong and GII.P16/GII.4, were obtained from GenBank with the accession numbers AAU95776.1, AAR97663.1, AAZ31396.2, ABQ63283.2, ADI45809.1, ALF12532.1, AGT95926.1, AFV99155.1 and BBA94050.1, respectively. The codon-optimized genes of the VP1 for these strains were cloned into the expression vector, which were then expressed by using the Hansenula polymorpha yeast expression platform constructed by our laboratory [27–29]. The recombinant

yeast cells were fermented for 4 days and collected by centrifugation with 3000 rpm for 5 minutes. After washing with crushing buffer, the collected cells were resuspended in the wash buffer. The supernatant was mechanically crushed three times by using the high-pressure cell cracker at 1300 bar, during which the temperature was maintained in the range of 2–8°C. The crushed crude solution was preliminary purified by using the aqueous two-phases extraction method, and then the expressed VP1s were obtained through further purification with the ion exchange chromatography method as described in our previous papers [27–29]. To confirm the successful assembly of the expressed VP1s into VLPs, the morphology of the particles was visualized by the negative stained transmission electron microscope (TEM). In the TEM observations, the purified VLP samples were dropped onto the carbon-coated copper carrier. After 5 minutes adsorption, the redundant samples on the carrier were removed with filter paper. The 2% phosphotungstic acid was added to the copper carrier for negative staining. The redundant staining solution was removed with filter paper and the carrier was dried by air at room temperature. Then the carrier was placed in the instrument and the morphology of the VLPs was observed by using the TEM.

## Binding assay of the VLPs with the synthetic HBGAs

The binding affinities of GII.4 variants with HBGAs were measured by the ELISA-based VLP-HBGA binding assay. In this study, the biotinylated polyacrylamide-conjugated carbohydrates (HBGA-PAA-biotin) were used, which were purchased from GlycoTech Corporation in the United States. A total of 9 types of PAA-biotin-conjugated HBGAs were tested in this study, which include H type 3 disaccharide; type A (tri), type B (tri), $Le^d$(H type 1), H type 2, $Le^a$ and $Le^x$ trisaccharides; and $Le^b$ and $Le^y$ tetrasaccharides. In the VLP-HBGA binding assay, the biotinylated HBGAs were diluted to 2.5 µg/ml with 0.1M sodium phosphate buffer (pH 6.4) and 0.25% bovine serum albumin, and added into the wells of the 96-well microtiter plate with 100 µl per well. After incubation at 25°C for 1 hour, the HBGAs were coated onto the wells of the plate. Then the wells were washed 4 times with 350 µl of 0.1M sodium phosphate at PH6.4 and blocked with 200 µl of 2.5% Blotto for 2 hours at 4°C. The purified GII.4 VLP samples were diluted to 7.2 µg/ml as the starting concentration, and a 3-fold serial dilution was carried out to obtain the VLP samples with different concentrations. The diluted VLP samples at various concentrations were added into the wells with 100 µl per well, which was incubated for 2 hours at 4°C. After washing four times with 350 µl 0.1M sodium phosphate at pH6.4, the bound VLPs were detected by using the HRP-conjugated rabbit anti-GII.4 NoV polyclonal antibody (PcAbs). The rabbit anti-GII.4 NoV antibody was diluted to the working concentration of 1:3200 and added into the wells with 100 µl per well. The plate was incubated for 1 hour at 4°C and washed 4 times with 350 µl 0.1M sodium phosphate at pH6.4. After that, the color reaction was performed for 5 minutes by adding the color development solution, which was then stopped by using the sulphuric acid. The optical density was measured at 450 nm and 630 nm ($OD_{450/630nm}$) with ELISA microplate reader.

## The characterization of the interactions between GII.4 NoV P domains and HBGAs by using MD simulations combined with MMGBSA method

In order to reveal the molecular mechanism for the evolution of the binding affinity of GII.4 NoV variants with HBGAs, the structure of P domain in complex with HBGA was constructed for different GII.4 strains and then the MD simulation combined with the MMGBSA method [30, 31] was used to investigate the interactions between the P domains and the HBGAs. In our simulations, five representative GII.4 strains (i.e., US 95/96, Farmington Hills, Sydney, Hong Kong and GII.P16/GII.4) and two HBGA types (i.e. A type and $Le^y$ type, which are

trisaccharide and tetrasaccharide, respectively) were selected to explore the mechanism for the evolution of the P domain-HBGA interactions. The crystal structures of the P domain of GII.4 Saga4 strain in complex with HBGA type A (PDB code: 4x07) and type Le$^y$ (4wze), respectively, have been resolved by Singh et al. [26]. In the present study, based on the crystal structure of 4x07, the complex structures of the P domain bound with type A HBGA for these different GII.4 strains were constructed by using the UCSF chimera software [35], in which the residue variations were introduced through the mutations of the corresponding residues. In the same way, the structures of the P domain in complex with type Le$^y$ HBGA for these various GII.4 strains were obtained based on the crystal structure of 4wze.

Then, these above constructed P domain-HBGA complex structures for different GII.4 strains were subjected to long-time MD simulations. For HBGA polysaccharides, the hydrogen atoms were added to the structure with UCSF chimera software, and the AM1-BCC method [36, 37] was employed to assign atomic charges to the HBGAs with Antechamber in Amber16 software [38, 39]. The preparation of the P domain-HBGA complex structures for MD simulations was performed by using LEAP module in AmberTools 16, and the ff14SB [40] and gaff force field parameters [41] were used for the NoV P domains and the HBGA carbohydrates, respectively. Then the simulated complex structure was solvated by TIP3P water molecules [42] in a cubic box, and the minimum distance between the surface of the solute and the edge of the box was set to 1.0 nm. The counter-ions were added into the box randomly to keep the simulation system being neutralized. After that, two-step energy minimizations were performed to remove the bad atomic contacts in the system. Firstly, the positions of the solute atoms were constraint with a force constant of 10 kcal/(molÅ$^2$), and 10000 steps minimization cycles, including 5000 steps steepest algorithm and 5000 steps conjugate gradient algorithm, were carried out to relax the water molecules. Then, the position restrains of the solute atoms were removed, and another 5000 steps steepest algorithm and 5000 steps conjugate gradient algorithm were performed to relax the whole system. After the energy minimizations, the system was heated from the 0K to 300k with position restraints of 10 kcal/molÅ$^2$ on the complex structure in a 200 ps constant volume (NVT) simulation. Following the heat process, five steps of constant pressure (NPT) equilibration simulations at the temperature of 300K were performed, in which the position restraints on the complex structure were gradually removed. For these five NPT equilibration simulations, the force constants of the position restraints were set to 5, 1, 0.25, 0.05 and 0 kcal/(molÅ$^2$), respectively, and each simulation was carried out for 200 ps. After the equilibration simulations, MD simulations without position restraints were performed for 20 ns. The MD simulations were run at the constant temperature of 300K and the constant pressure of 1 atm, in which the Berendsen temperature bath was used to regulate the temperature of the system and the SHAKE algorithm was used to constrain the bond length in the system. The cutoff for the non-bonded interactions was set to 10 Å and the time step in the simulations was set to 2 fs.

Based on the simulation trajectories, the molecular mechanics and generalized born surface area (MMGBSA) method was used to calculate the binding free energy between the NoV P domain and the HBGA molecule [31]. In the MMGBSA calculations, 100 snapshots were extracted from the last 10 ns trajectory for each MD simulations to compute the binding free energy, and then the relative binding affinity of HBGAs bound with different GII.4 NoV strains were compared to study the evolution of the NoV-HBGA interactions. All the MMGBSA calculations were performed by using the single-trajectory method with AMBER 16 program [31, 43]. Besides that, in order to investigate the molecular mechanism behind the evolution of NoV-HBGA interactions, the per-residue contribution to the binding free energy, the conformational rearrangements in the loop region around the receptor-binding pocket, as well as the hydrogen bonds interactions around the binding pocket were analyzed in detail.

The numerical data that were used to generate all figures are included in S1 Data in the Supporting information.

## Supporting information

**S1 Fig. The time-evolution of RMSD for the MD simulations of the various GII.4 NoV P domains complexed with the receptor HBGAs.** (A) The RMSD plots for the P domains of the US95/96 (black), Farmington Hills (red), Sydney (blue), Hong Kong (green) and GII.P16/GII.4 (purple) strains, respectively, in complex with type A HBGA. (B) The RMSD plots for the P domains of the US95/96 (black), Farmington Hills (red), Sydney (blue), Hong Kong (green) and GII.P16/GII.4 (purple) strains, respectively, in complex with type Le$^y$ HBGA. (TIF)

**S2 Fig. The residue contacts in loop-2 analyzed by using the "Find Clashes/Contacts" tool of the UCSF Chimera software for the Hong Kong and the GII.P16/GII.4 strains of GII.4 NoV in complex with type A HBGA, respectively.** (A) The residue contacts in loop-2 for the Hong Kong strain. (B) The residue contacts in loop-2 for the GII.P16/GII.4 strain. (TIF)

**S3 Fig. The residue contacts in loop-2 analyzed by using the "Find Clashes/Contacts" tool of the UCSF Chimera software for the Hong Kong and the GII.P16/GII.4 strains of GII.4 NoV complexed with type Le$^y$ HBGA, respectively.** (A) The residue contacts in loop-2 for the Hong Kong strain. (B) The residue contacts in loop-2 for the GII.P16/GII.4 strain. (TIF)

**S1 Data. The numerical data to generate Figure panels 2A, 2B, 2C, 2D, 2E, 2F, 2G, 2H, 2I, 4B, 5A, 5B, 5C, 5D, S1A, S1B.** (XLSX)

## Author Contributions

**Conceptualization:** Ji Guo Su, Qi Ming Li.

**Data curation:** Wei Bu Wang, Fang Tang.

**Formal analysis:** Wei Bu Wang, Jun Wei Hou, Fang Tang, Xue Feng Zhang, Li Fang Du.

**Funding acquisition:** Qi Ming Li.

**Investigation:** Yu Liang, Wei Bu Wang, Jing Zhang, Jun Wei Hou, Fang Tang, Xue Feng Zhang, Li Fang Du.

**Methodology:** Yu Liang, Jing Zhang.

**Project administration:** Jing Zhang, Qi Ming Li.

**Software:** Wei Bu Wang.

**Supervision:** Jing Zhang, Ji Guo Su, Qi Ming Li.

**Validation:** Fang Tang, Xue Feng Zhang, Li Fang Du.

**Writing – original draft:** Ji Guo Su.

**Writing – review & editing:** Qi Ming Li.

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
