## [Decision Letter · Decision Letter 0]

21 Apr 2021

Dear Prof. Su,

Thank you very much for submitting your manuscript "Evolution of the interactions between GII.4 noroviruses and histo-blood group antigens: Insights from experimental and computational studies" for consideration at PLOS Pathogens. As with all papers reviewed by the journal, your manuscript was reviewed by members of the editorial board and by several independent reviewers. The reviewers appreciated the attention to an important topic. Based on the reviews, we are likely to accept this manuscript for publication, providing that you modify the manuscript according to the review recommendations.

This study is an additional analysis of the known factor that is associated with susceptibility and infection of humans and noroviruses. They use yeast expression to prepare the capsids of several strains from a recent common lineage (GII.4) of human noroviruses that has become widespread around the world. They perform binding studies to the different blood group glycans, showing increasing binding over time of evolution of the viruses. They model the docking of the glycans into the structures of the viruses based on the sequences and known structures of related viruses binding to the different blood group antigens. The data is similar to previous studies, but adds additional approaches as well as details from the more recent viruses. That data supports the general connection of the affinity of binding to the increased spread of the viruses, but there is no direct connection back to the disease in vivo, and data is mostly suggestive. The manuscript is clear and well presented, and the data appear to be fine as far as they go.

Sincerely,

Colin Parrish, Ph.D.

Editorial Advisor

PLOS Pathogens

Raul Andino

Section Editor

PLOS Pathogens

Kasturi Haldar

Editor-in-Chief

PLOS Pathogens

orcid.org/0000-0001-5065-158X

Michael Malim

Editor-in-Chief

PLOS Pathogens

orcid.org/0000-0002-7699-2064

This study is an additional analysis of the known factor that is associated with susceptibility and infection of humans and noroviruses. They use yeast expression to prepare the capsids of several strains from a recent common lineage (GII.4) of human noroviruses that has become widespread around the world. They perform binding studies to the different blood group glycans, showing increasing binding over time of evolution of the viruses. They model the docking of the glycans into the structures of the viruses based on the sequences and known structures of related viruses binding to the different blood group antigens. The data is similar to previous studies, but adds additional approaches as well as details from the more recent viruses. That data supports the general connection of the affinity of binding to the increased spread of the viruses, but there is no direct connection back to the disease in vivo, and data is mostly suggestive. The manuscript is clear and well presented, and the data appear to be fine as far as they go.

Reviewer Comments (if any, and for reference):

Reviewer's Responses to Questions

**Part I - Summary**

Reviewer #1: In this article, the authors study the interactions between different types of histo-blood antigens (HBGAs) and virus-like particles of nine representative genotype II.4 strains of Norovirus. This work aims to understand binding affinity evolution at the structural (atomic) level. Indeed, the biding activity of HBGAs increases along the evolution of genotype II.4. The study combines both experimental and computational methods. In particular, the authors used molecular dynamics and molecular mechanic/generalized born surface area calculations to interpret enzyme linked immunosorbent assays. This allows the authors to interpret mutations consequences for binding affinity.

It is a very interesting work. The results are clear and well described. The methods are rigorously applied with correct interpretation.

This work is suitable for the audience of Plos Pathogens.

**Part II – Major Issues: Key Experiments Required for Acceptance**

Reviewer #1: (No Response)

**Part III – Minor Issues: Editorial and Data Presentation Modifications**

Reviewer #1: Minor point: the authors indicated p25 line 531 that 100 snapshots were used to compute the binding free energy for each simulation. However, in figure 4B, the curves presented do not take account of the statistics. I guess the means are plotted but the authors should show error bars.

It would also be informative to present the RMSD for each molecular dynamic simulations in the SI.

PLOS authors have the option to publish the peer review history of their article (what does this mean?). If published, this will include your full peer review and any attached files.

Reviewer #1: No

Figure Files:

Data Requirements:

Reproducibility:

References:

---

## [Editor Report · Decision Letter 1]

23 Jun 2021

Dear Prof. Su,

We are pleased to inform you that your manuscript 'Evolution of the interactions between GII.4 noroviruses and histo-blood group antigens: Insights from experimental and computational studies' has been provisionally accepted for publication in PLOS Pathogens.

Best regards,

Raul Andino

Section Editor

PLOS Pathogens

Raul Andino

Section Editor

PLOS Pathogens

Kasturi Haldar

Editor-in-Chief

PLOS Pathogens

orcid.org/0000-0001-5065-158X

Michael Malim

Editor-in-Chief

PLOS Pathogens

orcid.org/0000-0002-7699-2064
---

## [Editor Report · Acceptance letter]

6 Jul 2021

Dear Prof. Su,

We are delighted to inform you that your manuscript, "Evolution of the interactions between GII.4 noroviruses and histo-blood group antigens: Insights from experimental and computational studies," has been formally accepted for publication in PLOS Pathogens.

Best regards,

Kasturi Haldar

Editor-in-Chief

PLOS Pathogens

orcid.org/0000-0001-5065-158X

Michael Malim

Editor-in-Chief

PLOS Pathogens

orcid.org/0000-0002-7699-2064